# Thermal Infrared Imaging-Based Affective Computing and Its Application to Facilitate Human Robot Interaction: A Review

**Chiara Filippini** [1,2,*], **David Perpetuini** [1], **Daniela Cardone** [1], **Antonio Maria Chiarelli** [1] **and Arcangelo Merla** [1,2]

1   Department of Neuroscience and Imaging, Institute for Advanced Biomedical Technologies, University G. D'Annunzio of Chieti-Pescara, Via Luigi Polacchi 13, 66100 Chieti, Italy; david.perpetuini@unich.it (D.P.); d.cardone@unich.it (D.C.); antonio.chiarelli@unich.it (A.M.C.); arcangelo.merla@unich.it (A.M.)
2   Next2U s.r.l., Via dei Peligni 137, 65127 Pescara, Italy
*   Correspondence: chiara.filippini@unich.it; Tel.: +39-0871-3556954

**Abstract:** Over recent years, robots are increasingly being employed in several aspects of modern society. Among others, social robots have the potential to benefit education, healthcare, and tourism. To achieve this purpose, robots should be able to engage humans, recognize users' emotions, and to some extent properly react and "behave" in a natural interaction. Most robotics applications primarily use visual information for emotion recognition, which is often based on facial expressions. However, the display of emotional states through facial expression is inherently a voluntary controlled process that is typical of human–human interaction. In fact, humans have not yet learned to use this channel when communicating with a robotic technology. Hence, there is an urgent need to exploit emotion information channels not directly controlled by humans, such as those that can be ascribed to physiological modulations. Thermal infrared imaging-based affective computing has the potential to be the solution to such an issue. It is a validated technology that allows the non-obtrusive monitoring of physiological parameters and from which it might be possible to infer affective states. This review is aimed to outline the advantages and the current research challenges of thermal imaging-based affective computing for human–robot interaction.

**Keywords:** human–robot interaction; thermal IR imaging; affective computing; social robots; emotion recognition

## 1. Introduction

Human–robot interaction (HRI) can vary depending on the specific robotic function. In fact, robots have been effectively used in the last decade in therapy and educational interventions, but they have been also used where human interaction is not required, such as for example in robotic vacuum cleaner applications. Even though the latter were not designed to interact socially and form bonds with humans, it was found that people often attribute human-like characteristics to their robotic technology and that they express attachment toward them [1]. This is in line with the assertion that humans have a natural tendency to anthropomorphize everything around them, including technology [2]. This goes especially with children, who "are unlikely to only use a robot as a tool, and they will undoubtedly have some sort of interaction that can be considered social" [3]. Therefore, it would be desirable that robots, which are designed to interact with adults and children, would be able to socially interact. To facilitate the interaction, robots should be easy to use, and they should be built to understand and correctly respond to different needs. As robots gain utility, and thereby influence on society, research in HRI is becoming increasingly important. HRI primarily deals with social robotics and includes



relevant aspects of robot action, perception, and cognition. The development of social robots focuses on the design of living machines that humans would perceive as realistic, effective, communicative, and cooperative [4]. For this purpose, social robots should be able to express, through their shapes and behaviors, a certain degree of "intelligence" [5]. This skill entails the whole set of social and perceptual abilities of the robot, delivering a human-like interaction.

However, on the other hand, the matter about the anthropomorphism of social robots is still strongly debated. Indeed, the inclination to anthropomorphize the existing technology could have negative consequences [6] and even lead to an unrecognized change in human relationships [7] or invasion of privacy [8]. Therefore, it would be prudent to distinguish cases in which the use of anthropomorphism can be encouraged and cases in which it would be better not to [9]. Darling [10] argues that anthropomorphic framing is desirable where it enhances the function of the technology. Especially, it should be encouraged for social robots and discouraged for robots that are not designed in an intrinsically social way [9,10]. The present review is focused on social robots and in this context, their anthropomorphism is aimed at facilitating the interaction with humans.

An important key to reproduce a human-like behavior and facilitate HRI is the understanding of human emotional reactions. In recent years, a great effort was devoted to endowing the robot with the capability of interpreting and adapting to humans' emotional state. Consequently, a core aspect of HRI is Affective Computing. The term Affective Computing was created over 10 years ago; it is defined as "computing that relates to, arises from, or deliberately influences emotion or other affective phenomena" [11]. Since then, the community developed different concepts, models, frameworks, and demo applications for affective systems that are able to recognize, interpret, process, and simulate human feelings and emotions. However, these activities have been currently limited to possibility studies, proof-of-concept prototypes, or demonstrators.

Human emotions are indeed manifested as visible changes in facial expressions, gestures, body movements, or voice tone [12]. Beyond these observable expressive channels, physiological modulations also occur and can be observed as modifications in blood pressure, heart rate, or electrodermal activity [13,14]. These modulations are controlled by the autonomic nervous system and are not directly visible to humans. The main approaches to objectively measure these emotional signs rely on the observation of the face, gestures or body posture, and on measurements of physiological parameters through contact sensors [15,16]. Visual observation has the advantage of being completely non-invasive and only moderately intrusive. In fact, whereas a camera is being considered an intrusion into privacy, people are inclined to accept and forget it once they get used to it and see a value in it [17]. Furthermore, the greatest advantage of using the visual domain for emotion recognition is that it relies on years of study and cutting-edge algorithms developed for face identification and facial expressions analysis. The disadvantage of the visual approach is that people are prone to avoid facial expressions when interacting with technical systems [18]. In fact, the facial display of emotional states characterizes the human-to-human interaction, and it was acquired through evolution to facilitate communication and to influence others' actions [19,20]. Since machines do not still respond to emotional expressions, humans have not yet learned to use this channel when communicating with a robot. On the contrary, measuring physiological parameters has the advantage of evaluating features that people cannot control or mask [21]. Therefore, they may deliver much more reliable data about the emotional symptoms than visual channels. Physiological parameters such as heart rate, skin temperature, and electrodermal activity are very simple to acquire and analyze. The drawback of working with physiological readings is that they are mostly obtained through contact sensors [22]. Direct contact with the person's skin requires the willingness to correctly wear the device. Novel wearable sensors have to meet various technological requirements (e.g., reliability, robustness, availability, and quality of data), which are often very difficult to obtain. Finally, the time required for sensors placing is not negligible.

In recent years, thermal InfraRed (IR) imaging has been used for Affective Computing [23–25], and it exploits the advantages of both visual and physiology measuring approaches, as well as overcoming their drawbacks. Thermal IR imaging-based Affective Computing is a breakthrough technology that

enables monitoring human physiological parameters and Autonomic Nervous System (ANS) activity in a non-contact manner and without subject's constraining [26]. Although thermal IR imaging has been widely used for human emotion recognitions [27–35], its application in HRI implies overcoming ambitious and entirely new challenges. The most important ones are real-time monitoring and real-life applications. To this end, the use of a mobile and ideally miniaturized technology is essential to bridge the gap between laboratory and real-world applications. In addition, since robots should be commercially available, they need to include low-cost technology, which implies facing further issues such as low resolution and signal-to-noise ratio.

In this survey, the importance of facilitating HRI in different aspect of everyday life is firstly highlighted. The state-of-the-art of affective state recognition through thermal IR imaging is briefly reviewed. Afterwards, the above-mentioned challenges are examined, indicating what has been done so far and suggesting further development to ensure its future efficient use in the field. Finally, the possible impact of this technology for HRI applications in infants, children, and adults is highlighted.

## 2. Study Organization and Search Processing Method

This study has been carried out as a systematic literature review based on the original guidelines as proposed by Kitchenham [36]. The work aims to highlight the positive impact that the thermal IR imaging technology can have on HRI application, facilitating and enhancing this interaction by recognizing the human affective state. For this purpose, the research questions (RQ) addressed by this study were:

RQ1. What is the broad social impact of a facilitated HRI? Or Why is facilitated HRI important?

RQ2. What are the scientific bases for thermal IR imaging as an affective state recognition technique?

RQ3. What are the limitations of current research?

RQ4. Has thermal IR imaging already been addressed in the HRI field?

One of the goals of this study was to make this review as inclusive and practical as possible. Therefore, the databases searched were both Scopus and Google Scholar. All the papers published in conferences and journals between 2000 and 2020 were considered. Papers published from the year 2000 were considered, since the word "Affective Computing" was coined by Picard that year [6], and "Affective Computing" started to be applied in the HRI field. For each research question, a search process was applied.

Concerning RQ1, the search was based on the words "facilitate" and "human–robot Interaction". In the Scopus database, the survey was set up by searching for those words within the following fields: article title, abstract, and keywords. The basic search generated 441 results. In the Scholar database, on the other hand, the advanced search can be performed either by searching (i) the entire text or (ii) the title only. Therefore, the search was based on "facilitate human–robot interaction" within the entire text. A total of 360 results were obtained from the Scholar survey. The review papers and all the papers that did not refer to a user study or that did not relate to HRI were excluded, which reduced the considered pool to 155 papers. Within those papers, 130 were related to Scopus research, and 105 were from Scholar with an overlap of 80 papers. The manual review process was adopted for the final exclusion; the papers' abstracts were scanned, and all the papers that did not report experimental applications of human interaction were discarded. A total of 18 papers were discussed in the review concerning these keywords. Among those papers, 13 papers resulted from both Scopus and Scholar research, while 5 were from Scholar only.

With respect to RQ2 and RQ3, the searched keywords were "thermal imaging" OR "IR imaging" OR "thermography" AND "emotion recognition" OR "Affective Computing" OR "emotion". In the Scopus database, those keywords were surveyed in fields such as article title, abstract, and keywords. Whereas in Scholar, the advanced survey was carried out by searching for "thermal imaging" OR "IR imaging" OR "thermography" with at least one of these words: "Affective Computing" OR "emotion", and the field searched was the entire text. The search generated 115 results in Scopus and 163 in Scholar with an overlap of 95 papers. The results were scanned through a manual review procedure focused

on the papers' abstracts, which aimed to identify whether the considered works reported on thermal IR imaging for human emotion recognition. Papers not related to it were excluded. The resulting papers were analyzed and grouped based on their experimental applications, strengths and limitations were indicated, and 46 were reported in the present work. Among those papers, 40 papers resulted from both Scopus and Scholar research, while 6 were from Scholar only.

Finally, as for the RQ4, the keywords "thermal imaging" OR "IR imaging" OR "thermography" AND "human–robot interaction" were searched. The fields checked and the procedure performed were the same reported in the previous RQs. The basic search generated 13 results in Scopus and 388 in Scholar. Ten papers that resulted from Scopus search were also found in the Scholar research outcome. Exclusion criteria regarded all the results that were not conference or journal papers actually related to thermal IR imaging applied in the HRI field. From the latter, 20 papers of thermal IR imaging-based Affective Computing were selected and included in this work. Among those papers, a subset of 2 papers was linked to both RQ2/RQ3 and RQ4; therefore, it was reported in both sections.

## 3. The Importance of Facilitating Human–Robot Interaction

Robotic technologies and HRI are being increasingly integrated in real-life contexts. Modern society creates ever more spaces where robot technology is intended to interact with people. Robots applications can range from education to communication, assistance, entertainment, healthcare, and tourism. Hence, there is a need to better understand how robotic technologies shape the social contexts in which they are used [37]. In this section, the importance of a natural and efficient HRI, in three major fields such as education, healthcare, and tourism, is deepened.

Economic and demographic factors drive the need for technological support in education. The growing number of students per class, the reduction of the school budget, and the demand for greater personalization of curricula for children with diverse needs are encouraging research on technology-based support for parents and teachers [38]. Hence, the efficacy of robots in education is of primary interest. Movellan et al. deployed a fully autonomous social robot in a nursery school's classrooms for a period of 2 weeks in order to study whether the robot could improve target vocabulary skills in toddlers (18–24 months age) [39]. The results showed that the vocabulary skills improved significantly; in fact, the number of learned words increased by 27% when compared to a matched set of control. Considerable educational benefits can also be obtained from a robot that takes on the role of a novice (i.e., a care-receiving robot), thus allowing the student to take on the role of the instructor, which generally improves self-confidence and learning outcomes. This phenomenon is known as learning by teaching. An example is the educational use of Pepper (Figure 1), which was designed to learn together with children at their home environment from a remote teacher [40].

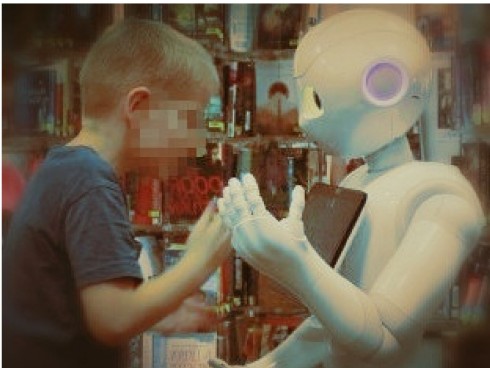

**Figure 1.** Pepper robot interaction with a child.

Social robots are also widely used in healthcare fields. Given the exponential growth of vulnerable populations (e.g., the elderly, children with developmental disabilities, and sick people), there is an increasing demand for social robots that are able to provide aids and entertainment for patients in the

hospital or at home. For instance, an important contribution can be provided by companion robots especially among sick people, in the mitigation of boredom, depression, isolation, and loneliness. In this perspective, Banks et al. explored the ability of a robotic dog (AIBO) to treat loneliness in elderly patients living in long-term care facilities (LTCF). Results demonstrated that LTCF residents showed a high level of attachment to AIBO, highlighting the capability of interactive robotic dogs to reduce loneliness [41].

Together with companion robots, therapy robots are also considered of high social impact. Therapy robots are generally employed to deliver treatment for people with physical and mental diseases, such as Autism Spectrum Disorder (ASD). A recent review reported statistics showing an annual increase in the number of children with ASD, starting from 1 out of 1000 children in 1970 up to 1 out of 59 children in 2018 [42]. Thus, the need for innovative care and proper attention for ASD children is compelling. Particularly, researches have shown that people suffering from Autism Spectrum Disorders (ASD) responded better to treatments involving robotic technology rather than treatments from human therapists [43]. Moreover, it was demonstrated that the use of robots in education helps ASD children to improve their abilities to handle social and sensory challenges at school environment and to better control the anxiety and stress [44,45]. Furthermore, Wilson et al. noted that one of the major barriers to the education of children with autism pertains to lack of knowledge, training, and specialized support staff as well as the lack of adequate resources for education and classroom size [46]. Therefore, the development of robots with advanced emotion recognition and HRI capabilities that can be used at school or at home is becoming essential in supporting ASD patients.

Finally, several applications in the literature confirmed the importance of HRI in tourism settings. In fact, with recent technological advancements in artificial intelligence (AI) and robotics, we see an increasing number of service robots entering tourism and hospitality contexts, including consumer-facing ones [47]. For instance, Niculescu et al. developed SARA (Singapore's Automated Responsive Assistant), a robotic virtual agent, to offer information and assistance to tourists, being able to detect the user's location on a map [48]. CLARA is a virtual restaurant recommendation system and conversational agent that provides tourists with information about sightseeing, restaurants, transportation, and general information about Singapore [49]. However, the adoption of service robots inevitably changes the nature of service experience. Unlike industrial robots whose performance metrics depend entirely on efficiency, the success of service robots depends on user satisfaction and, consequently on the degree of empathy and natural interaction. Tussyadiah et al. focused on consumer's evaluation of hotel service robots. In their study, they surveyed consumer response to two types of robots, NAO and Relay. The results revealed that consumer intention to accept hotel service robots is influenced by their interaction with the robot, dimensions of anthropomorphism, perceived intelligence and security [47]. The same conclusion was drawn from Kervenoael et al. in [50], where they stated that empathy from service robots has a positive and significant effect on the intention to use a robot. Empathy is meant as the robot's ability to understand or feel what another person is experiencing from within their frame of reference, i.e., the robot's ability to recognize emotion and to respond it in an appropriate way. In conclusion, beyond the requirement of well-programmed social robots, in order to cater to specific consumers or interlocutor's needs, they must incorporate a seamless integration of safe and reliable service that includes courtesy and inspires trust.

With sophisticated robots or artificial agents becoming ever more ubiquitous in daily life, their appropriate use is crucial. In this section, we provided examples of their positive impact on fields such as education, healthcare, and tourism. Research on HRI requires contributions and expertise from heterogeneous disciplines, including engineering and artificial intelligence. In fact, to ensure a natural HRI, social robots need to recognize human emotions, respond adequately to them, understand and generate natural language, have reasoning skills, plan actions, and execute movements in line with what is required by the specific context or situation [51]. A good extent of the effort can be ascribed to emotion recognition, which currently requires exploring fields of facial expression analysis and speech processing, which are anything but trivial tasks. Relying on emotion recognition through

non-obtrusive physiological sensing, thermal IR imaging-based affective computing can be a way to avoid some of those challenges. Although this technique is already used in the field of emotion recognition, methodological considerations are required to make it suitable for HRI applications.

## 4. Affective States Recognition through Thermal IR Imaging

A considerable number of studies have explored the use of thermal IR imaging for classifying affective states and human emotions. Those studies were based on measuring a person's physiological cues that are considered related to affective states. Indeed, the observations of affective nature derive primarily from muscular activity, subcutaneous blood flow, perspiration patterns in specific body parts, as well as from changing in breathing or heart rate, which are all phenomena that are controlled by our ANS. Measuring facial cutaneous temperature and assessing both its topographic and temporal distribution can provide insights about the person's autonomic activity. This depends on the ANS's role in the human body's thermal homeostasis and in the regulation of physiological responses to emotional stimuli [29]. Alert, anxiety, frustration responses, and other affective states determine the redistribution of the blood in the vessels through vasodilation or vasoconstriction phenomena which are regulated by ANS. These phenomena can be captured by an IR thermal camera through changes in the IR emissivity of the skin. Vasoconstriction implies a decrease in temperatures of the Region Of Interest (ROI). On the other hand, vasodilatation is the cause of heating. Vasomotor processes can be identified and monitored over time because they produce a thermal variation of the skin and they can be characterized by simple metrics such as temperature difference between data at two temporal points. For this reason, most researchers have focused on studying the relationship between thermal directional changes (i.e., temperature drop and rise) of specific skin areas in relation to psychophysiological states [28,31,52,53]. As a body area of interest, the human face is considered of particular importance since it can be easily recorded and it is naturally exposed to social stimuli. Regions of the face extensively characterized are the nose or nose tip, the glabella (area associated with the corrugator muscle), the periorbital area, forehead, and the orbicularis oculi (surrounding the eyes), as well as the maxillary area or the upper lip (perinasal) [24,28,54]. Partially evaluated regions were cheeks, carotid, eyes, fingers, and lips. An exhaustive review on this topic by Ioannou et al. summarized the emotions, the observed regions, and the direction of the average temperature changes in those regions (Table 1) [55].

**Table 1.** Overview of the temperature variations direction, as estimated through thermal IR imaging, for each emotion in the different regions considered. The table is adapted from [55].

| | Stress | Fear | Startle | Sexual Arousal | Anxiety | Joy | Pain | Guilt |
|---|---|---|---|---|---|---|---|---|
| **Nose** | ↓ | ↓ | | ↑ | | ↓ | | ↓ |
| **Cheeks** | | | ↓ | | | | | |
| **Periorbital** | | | ↑ | ↑ | ↑ | | | |
| **Supraorbital** | | | ↑ | | ↑ | | | |
| **Forehead** | ↓ ↑ | ↓ | | ↑ | ↑ | | ↓ | |
| **Maxillary** | ↓ | ↓ | ↓ | | | | ↓ | ↓ |
| **Neck-carotid** | | | ↑ | | | | | |
| **Finger/palm** | | | | | | | ↓ | |
| **Lips/mouth** | | | | ↑ | | | | |

Similar results were found in a recent study by Cruz-Albarran et al., where thermal IR imaging was used during the emotions induction process to quantify temperature changes that occurred on different ROIs of the face [56]. The authors were able to classify the emotions relying on regional temperatures with an accuracy of 89.9%. The induced emotions were joy, disgust, fear, anger, and

sadness. The examined ROIs were nose, cheeks, forehead, and maxillary area, which are depicted in Figure 2. Among all the ROIs, the nose and maxillary area were the most responsive to emotional stimuli, as they showed a significant change in temperature in all the induced emotions. The forehead temperature changed during sadness, anger, and fear, while the temperature of the cheeks changed during disgust and sadness.

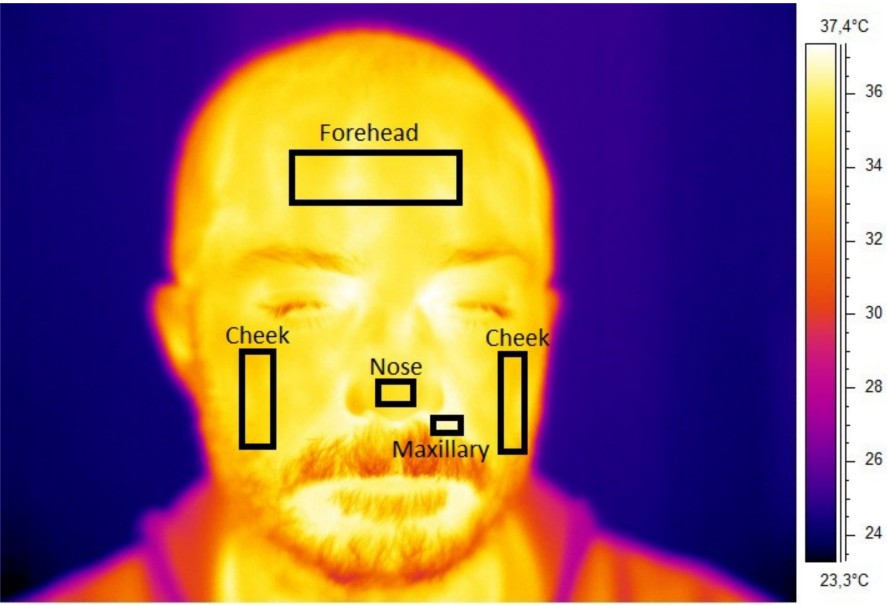

**Figure 2.** Thermal IR images with marked ROIs (black rectangles).

Emotion identification through thermal IR imaging was also employed in studies on children. Such applications may receive the greatest benefit from the technology thanks to the ecological nature of this technique and the difficulties associated with measuring children with skin-located sensing. Goulart et al. proposed an experimental design to identify five emotional states (disgust, fear, happiness, sadness, and surprise) evoked in children between 7 and 11 years old [57]. The forehead, the tip of nose, the cheeks, the chin, the periorbital and the perinasal regions were chosen to extract the affective information. Each thermal frame was processed by segmenting the ROIs and evaluating the ROIs' temperature mean, variance, and median values. Then, a linear discriminant analysis was used as a classifier. High accuracy (higher than 85%) was obtained for the classification of the five emotions, thus resulting in a robust method to identify quantitative patterns for emotion recognition in children. Temperature decrease was detected in the forehead region during disgust and surprise; in the periorbital region during happiness, sadness, and surprise; in the perinasal region during disgust and happiness; in the chin during surprise, happiness, and sadness; and finally, in the nose during disgust, fear, and happiness. The temperature increase was detected in the left cheek region for all emotions and in the nose tip during surprise.

Beyond the basic emotions, thermal IR imaging has been used to characterize the two dimensions of emotions, such as valence (pleasant versus unpleasant) and arousal (low versus high). Emotions dimensions are a crucial aspect in the affective research field, on which most of the studies on emotions recognition are based. The most commonly used models representing emotions dimension in HRI are the Pleasure, Arousal, Dominance (PAD) emotional state model [58] and the circumplex model of affect [59]. The PA dimensions of PAD were developed into the circumplex model, which indeed assume that any emotion might be described with two continuous dimensions of valence and arousal [60]. The valence dimension indicates whether the subject's current emotional state is positive or negative. Arousal, on the other hand, indicates whether the subject is responsive or not, at that given moment and for that given stimulus, and how active he/she is. In particular, the theory of the dimension of the emotions proposes that the emotional states are not discrete categories but rather a result of varying

degrees of their dimensions. A graphical representation of the circumplex model of affect developed by Russel is reported in Figure 3. For example, joy is characterized as the product of strong activation in the neural systems associated with positive valence or pleasure together with moderate activation in the neural systems associated with arousal (i.e., low arousal). Emotions other than joy likewise arise from the same two-dimensional systems but differ in the degree or extent of activation. This allows characterizing also complex emotions such as love or happiness other than the basic ones. The analysis of the emotion recognition solutions reveals that there is no one commonly accepted standard model for emotion representation. The dimensional adaptation of Ekman's six basic emotions and the circumplex or PAD model are the ones widely adopted in emotion recognition solutions [61].

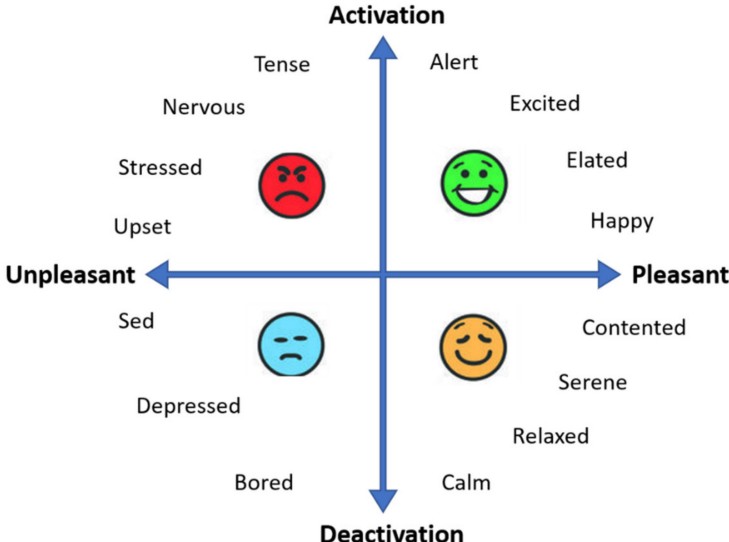

**Figure 3.** Graphical representation of the circumplex model of affect with the horizontal axis representing the valence dimension and the vertical axis representing the arousal dimension. Adapted from [60].

Recent studies explored temperature changes associated with different degrees of valence and arousal. For instance, Salazar-Lopez et al. studied the relation between changes in temperature of the subject's face and valence or arousal dimensions [53]. They used pictures from the International Affective Picture System (IAPS), which is widely used in studies of emotion recognition and characterized along the two dimensions [62]. The analyzed ROIs were the forehead (left and right sides), the tip of the nose, the cheeks, the eyes, and the mouth regions. Significant differences in temperature were found only on the tip of the nose. The results showed that high arousal images elicited temperature increases on the tip of the nose, while low arousal images led to temperature increases for pleasant images (i.e., positive valence) and decreases for unpleasant ones (i.e., negative valence). Contrasting results were indeed found by Kosonogov et al. [63]. The authors found no significant temperature differences along the valence dimension of the emotions (i.e., pleasant and unpleasant emotions). Besides, an activation effect of emotional pictures was found on the amplitude and latency of nasal thermal responses: the more arousing the pictures, the faster and larger the thermal responses. The only evaluated region was the tip of the nose. The relevance of the emotional arousal in causing changes in the nose tip temperature is supported and demonstrated in different studies, such as in Diaz-Piedra et al., where a direct relationship was found between reduced levels of arousal and nasal skin temperature [64], and confirmed in Bando et al. [65].

Moreover, Pavlidis et al. proposed quantifying stress through thermal IR imaging [54]. Stress in the valence–arousal space is identified as negative valence and high arousal [66]. Pavlidis et al. found that high stressful situations resulting in the cooling of the area around the nose tip [54]. Parallel results were found in children, Ioannou et al. obtained, in their study of guilt in children (age 39–42 months), in which the higher the distress signs, the higher the decrease in nose temperature [67]. The sense of guilt

was induced through the "mishap paradigm" in which children were led to believe they had broken the experimenter's favorite toy (Figure 4). The temperature of the nasal area decreased following the "mishap" condition, suggesting a sympathetic activation and peripheral nasal vasoconstriction, and it increased after soothing due to a parasympathetic activation.

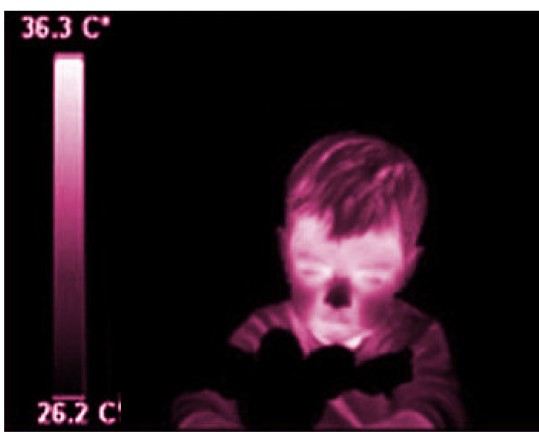

**Figure 4.** Child facial temperature during the mishap condition. Adapted from [68].

Vasomotor process, subcutaneous blood flow, and perspiration patterns are not the only physiological measures that can be detected through thermal IR imaging. In fact, the physiological signature of the autonomic system activation such as heart and breathing rate variation can be also monitored [69–72]. These two parameters are measurable with thermal IR imaging by positioning a ROI over a superficial vessel or over the nostrils respectively and by monitoring the ROI average temperature over time. Whereas the latter is easily detectable, because the thermal difference between the expired and inspired air is easily appreciable (approximately 0.5 °C), the modulation of temperature caused by the pulsation of blood in the vessel is not. Therefore, a series of algorithms have been developed for the estimation of the heart rate through thermal imaging [69]. Cross et al. used thermal IR imaging to detect physiological indicators of stress in the adult population by analyzing the respiration and heart rate variation during the performance of mental and physical tasks [26]. Temperature variation over time was recorded on the nose tip and regions near superficial arteries to detect respiration and heart rates, respectively. The results showed that the accuracy in the physical versus psychological stressors classification was greater than 90%, and the heart and respiration rate were accurately detected by thermal IR imaging. Whereas the evaluation of the heart rate through thermal IR imaging is not common in the literature, the respiration rate assessment through thermal IR imaging has been tested in different settings, including sleep monitoring [73], neonatal care [74], and driver's drowsiness monitoring [75].

## 5. Limits of Current Thermal IR Imaging for HRI Applications

Studies reviewed in this section revealed thermal IR imaging ability to monitor physiological signs and affective states. Although they have sometimes shown incongruent results (e.g., nose tip temperature did not significantly change along the valence dimension of the emotions in Kosonogov et al. [63], whilst it did change in Salazar-López et al. [53]), these findings open exciting prospects for affective computing. One of the causes of inconsistency could be that a single discrete metric is maybe not always sufficient, since it could be susceptible to the complex physiological mechanisms [76]. In addition, there could be interaction effects between affective states. Nonetheless, all the affective states analyzed have the capability to induce ROI temperature variations. However, an important consideration is that all these studies were conducted using high-end thermal camera and performed in controlled laboratory settings. The thermal systems mostly used in literature are FLIR (Wilsonville, OR, USA) A655sc with 640 × 480 spatial resolution, a 50 Hz sampling rate, and <0.03 °C

thermal sensitivity and a FLIR A325sc with 320 × 240 spatial resolution, 60 Hz sampling rate, and <0.05 °C thermal sensitivity. In addition, all the observations reported were made in a climate-controlled room according to the International Academy of Thermology (IACT) guidelines [77]. In fact, IACT guidelines indicated that when performing a thermal IR imaging measurement, it is mandatory to control the temperature and humidity of the experimental room. They suggested a temperature range of 18–23 °C and a controlled humidity range between 40% and 70%. No direct ventilation on the subject and no direct sunlight (no windows or with curtains or blinds) is also recommended during the experimental measures. In conclusion, by analyzing the studies reported in this section, two main constraints were identified that are not suitable for HRI applications. Those are (1) the use of high-end and sized thermal imaging systems and (2) the circumstances in which the studies were conducted i.e., in restricted laboratory settings. On the other hand, HRI applications require daily life scenarios, eventually suitable for outdoor use, and technology embedded in commercial social robots, i.e., low-cost miniaturized sensors. In Sections 6 and 7, those limits are addressed in order to highlight the new improvements developed in recent studies. A special emphasis was placed on these two sections as they deal with crucial aspects for the use of thermal IR imaging in the HRI field. Finally, the last section focuses on the actual state of the art of thermal IR imaging-based affective computing applications.

## 6. Mobile Thermal IR Imaging

The relevant spread of thermal IR technology, together with the miniaturization of IR detectors, induced manufacturers to produce portable thermographic systems, i.e., mobile and low-cost thermal IR imaging devices. One of the first companies to commercialize mobile thermal devices was FLIR with systems such as the FLIR ONE Pro, 160 × 120 spatial resolution, an approximately 8.7 Hz sampling rate, 0.15 °C thermal sensitivity, and dimensions of 68 × 34 × 14 mm$^3$, or FLIR Lepton, 80 × 160 spatial resolution, an approximately 8.7 Hz sampling rate, <0.05 °C thermal sensitivity, and dimensions of 11.8 × 12.7 × 7.2 mm$^3$. FLIR ONE was designed to be integrated on mobile phones. Another mobile thermal system designed to be integrated on a mobile phone is the Therm-App system developed by Opgal manufacturer; it has 384 × 288 spatial resolution, an approximately 9 Hz sampling rate, approximately 0.07 °C thermal sensitivity, and dimensions of 55 × 65 × 40 mm$^3$. Market research has shown that the SmartIR640 mobile thermal system, manufactured by Device aLab (640 × 480 spatial resolution, 30 Hz sampling rate, <0.05 °C thermal sensitivity, and dimensions of 27 × 27 × 18 mm$^3$) is also a valid solution, but it is not yet used for research projects.

Despite the relatively low-quality thermal imaging outputs of a mobile thermal system, this technology could help bridge the gap between the findings from highly constrained laboratory environments and wild real-world applications. Indeed, its portability (e.g., small size and low computational resource requirement) allows the camera not only to be easily attached to a mobile phone but also to be integrated in a social robot head. Recent studies have started to explore mobile thermal IR imaging for affect recognition tasks, especially focused on stress monitoring [78–81]. Cho et al. proposed a system consisting of a smartphone camera-based PhotoPlethysmoGraphy (PPG) and a low-cost thermal camera added to the smartphone, which was designed to continuously monitor the subject's mental stress [78]. By analyzing the nose tip temperature and the blood volume pulse through PPG [82,83], they were able to classify the stress with an accuracy of 78.33%, which is comparable to the state-of-the-art stress recognition methods. The employed mobile thermal camera was the FLIR ONE. The study was conducted in a quiet laboratory room with no distractions. Another study by the same authors included the mobile thermal camera as a standalone system to monitor mental stress [80]. The authors proposed a novel low-cost non-contact thermal IR imaging-based stress recognition system that relayed on the breathing dynamic patterns analysis. In fact, since breathing is an important vital process controlled by the ANS, its pattern monitoring can be informative of a person's mental and physical condition. Results showed a classification accuracy of 84.59% for a binary classification (i.e., no-stress, stress) and an accuracy of 56.52% for multi-class classification (i.e.,

none, low, high-level stress). The breathing signal was recovered by tracking the person's nostril area. Then, the extracted breathing signal was converted in a two-dimensional spectrogram by stacking the Power Spectral Density (PSD) vector of a short-time-window respiration signal over time. Since the PSD function handles the short-time autocorrelation that identifies similarities between neighboring signal patterns, it can be used to examine respiration variations in a short period [65]. The breathing signal was also investigated through the use of a mobile thermal camera by Ruminski et al., who embedded such a camera in smart glasses [84]. Basu et al. instead used a mobile thermal system for the challenging purpose of classifying personality (psychoticism, extraversion, and neuroticism) [81]. The proposed system classified the emotional state using an information fusion of thermal and visible images. A blood flow perfusion model was used to obtain discriminating eigenfeatures from the thermal system. Then, these eigenfeatures were fused with those of visible images and classified. The blood perfusion model was obtained by analyzing the thermogram of the entire face and using Pennes' bioheat equation. The classification performance reached 87.87%.

Summarizing, mobile thermal IR imaging can provide high levels of flexibility and suitability for recovering physiological signatures such as the breathing signal and recognizing a person's affective states. However, a key challenge on the use of a mobile thermal system is related to the low quality of the output signal due to the low thermal and spatial resolution of the imaging system. The low spatial resolution can be easily addressed by bringing the thermal camera closer to the region of interest, but this is not always possible. An interesting method to overcome such an issue is presented in Cho et al. [85]. The authors proposed the *Thermal Gradient Flow* and *Thermal Voxel Integration* algorithms. *Thermal Gradient Flow* was mainly based on building thermal-gradient magnitude maps for enhancing the boundary around the region of interest, which in turn contributes to making the system robust to motion artifacts in presence of low-resolution images. Instead, the *Thermal Voxel Integration* consisted of a projection of a 2D thermal matrix onto a 3D space by taking a unit thermal element as a thermal voxel. This method was applied for breathing signal analysis application, and it resulted in producing a higher quality of breathing patterns.

## 7. Thermal IR Imaging-Based Affective Computing Outside Laboratory Settings

The market entry of smaller and low-cost thermal cameras is paving the way for thermal IR imaging applications outside laboratory environments. However, only few studies have been conducted. One example is reported by Goulart et al., who proposed a camera system composed of thermal and visible cameras for emotion recognition in children [86]. The camera system was attached to the head of a social robot, and the experiment was conducted in a room within the children's school environment. The room temperature was kept between 20 and 24 °C, using a constant luminous intensity. To capture thermal variation in the children's faces, they used the mobile thermal camera Therm-App. A similar set-up has been reported in Filippini et al. [87]. The authors installed a mobile thermal IR imaging system, the FLIR ONE, on the head of a social robot. The study was conducted in a primary school. Filippini and Goulart's studies are described in detail in the next section. Although both experiments were performed outside a laboratory setting, they still had constraints that are not adaptable to all real-life applications, such as the need to maintain a stable temperature, which is not possible in open air contexts or applications. Instead, Cho et al. conducted one of their experiments in unconstrained settings with varying thermal dynamic range scenes (i.e., indoor and outdoor physical activity). The experiment was aimed to monitoring a person's thermal signatures while walking [85].

Concerning the employment of thermal mobile system for affective computing purposes, one of the most compelling challenges for real-world environment is to ensure reliable thermal tracking of the chosen ROIs. This is due to dynamic changes in ambient temperature that affect the skin and can cause an inconsistent thermal signal coupled with the low resolution of the low-cost thermal system. This aspect makes it difficult to track ROIs automatically. Moreover, applications in real-world scenarios require real-time responses from the sensors of interest. Hence, an automatic recording of thermal IR imaging data and real-time processing is required. In this regard, signal processing techniques need to

be chosen based on their efficiency in terms of computational load to allow acceptable performance for real-time processing. In Goulart et al. and Filippini et al., the tracking algorithm in thermal images relied on the visible images [86,87]. Thereby, the first phase consisted on the camera calibration done through a synchronous acquisition between visual and thermal images and using a checkerboard whose details were clearly detectable by both the visible spectrum and the thermal camera. After the calibration, the face detection and ROIs localization were performed on the visible image, since a state-of-the-art computer vision algorithm can be used in the visible field, and then the ROI coordinates in the visible images were converted to those of the thermal image. In Goulart et al., the Viola–Jones algorithm was used for visible ROIs detection, and then these ROIs were transferred to the corresponding thermal camera frame through a homography matrix [86]. In Filippini et al., the authors used an object detector based on the histogram of oriented gradients (HOG) for face localization in visible images [87]. The extraction of landmarks was based, instead, on a regression tree ensemble algorithm. An average of 82.75% of the total amount of the frames were correctly tracked as a result. However, it is important to mention that a further improvement would be to develop a real-time tracker, based on the only IR videos, acquired by low-resolution thermal cameras, to avoid problems due to a low-light environment and to infer the psychophysiology state of the human interlocutor. An attempt in this direction was proposed in Cho et al., where the authors were interested in tracking the physiological signal related to the only breathing process, focusing on the nostril region [85]. The approach they proposed was intended to compensate for the effects of variation in ambient temperature and movement artifacts, and it was named the "Optimal Quantization" method. The quantization process itself is the process of translating from a continuous temperature value to its digital color-mapped equivalent. This method consisted in adaptively quantizing the thermal distribution sequences by finding a thermal range of interest that contains the whole facial temperature distributions for every single frame. Despite the enhancing approach, this cannot cover all possible scenarios such as contexts of high humidity or severe temperature condition. Nonetheless, the further development of automatic ROI tracking on thermal images in entirely mobile and ubiquitous situations is required.

## 8. Thermal IR Imaging-Based Affective Computing in HRI

For decades, a growing interest on the possibility of developing intelligent machines that engage in social interaction has been observed. Many researchers, also in the field of thermal IR imaging, have been enthralled with the HRI appeal and with the possibility of developing robots that are capable of social interactions with humans. The application of thermal IR imaging in this field was firstly investigated by Merla in 2014 [88]. Since then, many studies have been conducted to validate the thermal IR imaging technique for emotion recognition analysis with the aim of applying this technique in the HRI field. However, only few applications have been actually implemented in this area. One of the first attempts was carried out by Sorostinean et al. [89]. In this study, the authors presented the design of a thermal IR imaging-based system mounted on a humanoid robot performing a contact-free measurement of temperature variations across people's faces in a stressful interaction. The results showed a statistically significant interaction between the distance and the gaze direction of the robot and the temperature variation of the nasal and peri-nasal region. This supported the fact that thermal imaging sensors can be successfully employed in embodying robots with physiological sensing capabilities to allow them to become aware of their effect on people, know about their preferences, and build a reactive behavior. Agrigoroaie et al. reported promising preliminary results in their attempt to determine if a person was trying to deceive a robot through the use of a mobile thermal camera [90]. Instead, Boccanfuso et al. evaluated the efficacy of the thermal IR imaging for detecting robot-elicited affective response compared to video-elicited affective response by tracking thermal changes in five ROIs on the subject's face [91]. They studied the interaction effects of condition (robot/video) and emotion (happy/angry) on individual facial ROIs. Although no interaction effects for most ROI temperature slopes were found, a strong, statistically significant effect of the interaction between condition and emotion when evaluating the temperature slope on the nose tip was observed. This result

confirms again the assumption that the nasal area is a salient region for emotion detection [32,67,92–95]. Recent studies included applications with more challenging populations such as infants and children. Scassellati et al. proposed the design of a unique dual-agent system that uses a virtual human and a physical robot to engage 6–12-month-old deaf infants in linguistic interactions. The system was endowed with a perception system that was capable of estimating infant attention and engagement through thermal imaging and eye tracking [96]. This study was part of a larger project designed to develop a system called RAVE (Robot AVatar thermal Enhanced language learning tool). RAVE is aimed to be an augmentative learning tool that can provide linguistic input, in particular visual language inputs, to facilitate language learning during one widely recognized critical developmental period for language (ages 6–12 months [97]) [96,98–101]. To this end, thermal IR imaging was used to determine the emotional arousal and attentional valence, providing new knowledge about when infants are most optimally "Ready to Learn", even before the onset of language production. This is prominent for infants who might not otherwise receive sufficient language exposure. Of particular concern are deaf babies, many of whom are born to parents who do not know a signed language [96].

Although these studies were very ambitious and fascinating, they were still carried out in constrained laboratory settings, probably implying a not entirely free interaction. To the state of the art, the only two studies performed so far that have concerned HRI applications in a out-of-laboratory context are those reported in Section 7 [86,87].

Goulart et al. used a mobile social robot called N-MARIA (New-Mobile Autonomous Robot for Interaction with Autistics), which was built in UFES/Brazil to assist children during social relationship rehabilitation [86]. In the interaction with the child, which lasted two minutes, the child was encouraged to make communication and tactile interaction with the robot. The robot was equipped with low-cost hardware (thermal and visible camera) used to give information about the emotional state of the child, which was related to five emotions (i.e., surprise, fear, disgust, happiness, and sadness). Results showed that the system was able to recognize those emotions, achieving 85.75% accuracy. Such accuracy was comparable with gold standard techniques in emotion recognition, such as facial expression analysis and speech tone analysis [102–106]. Filippini et al. used the social robot "Mio Amico" robot, produced by ©Liscianigiochi, in which the mobile thermal system FLIR ONE (which includes a thermal and a visible camera) was installed on the head of the robot [87]. The study aimed to endow the robot with the capability of real-time assessment of the interlocutor's state of engagement (positive, neutral, and negative emotional engagement). During the interaction between the robot and the child, the robot could either tell a fairy tale or sing a song. At the end of the fairy tale or the song, the robot asked the child if he/she liked the fairy tale/song and if he wanted to listen to another one. Based on the child's answer, the robot could choose the next action. The engagement state of the infant was classified by analyzing the child's thermal modulation using a low computational processing pipeline. Figure 5 summarizes in a–c the processing pipeline for the interlocutor's state of engagement identification and (d) the child–robot interaction. The accuracy reached was 70%. Although this study presented a lower level of accuracy compared to Goulart et al. [86], it is worth mentioning that the estimation of the level of engagement can be considered a hard task, since it represents a complex emotion (combination of the basic ones), and it has been poorly investigated in the literature. Besides, the study reported in Filippini et al. [87] represents the first and unique study in which the robot could actually change its activities based on the child affective state, opening the way to a bidirectional interaction.

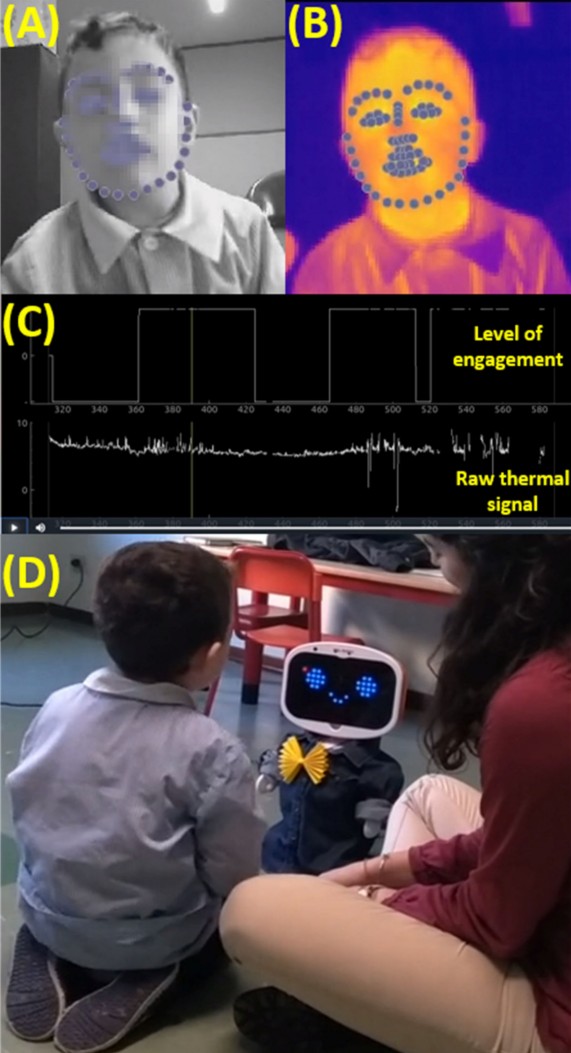

**Figure 5.** (**A**–**C**) Processing pipeline. (**A**) The first phase relied on the visible image to detect the child's face and locate specific facial landmarks. (**B**) The corresponding landmarks on the thermal image were then obtained thanks to a previous optical calibration procedure. (**C**) The thermal signal is extracted from the nose tip region and processed in order to obtain the subject's level of engagement. (**D**) The child–robot interactions are guided by the robot's understanding of the child's level of engagement.

## 9. Thermal IR Imaging-Based Affective Computing in Intelligent Systems Such as Driver-Assistance Systems or Autonomous Vehicles

Besides humanoid robots, systems such as driver-assistance systems or autonomous vehicles can also be classified as "robots" due to their intelligent behaviors. Indeed, driving functions are becoming increasingly automated; consequently, motorists could run the risk of being cognitively removed from the driving process. Thermal IR imaging was demonstrated to be a valid technique for assessing variation in cognitive load [52,107,108]. Most of all, it could enable sensing the real-time state of motorists non-invasively i.e., without disrupting driving-related tasks and, unlike RGB cameras, independently from external light conditions [109]. Studies of thermal IR imaging in driving contexts stated that a rise in mental workload leads to an increase in the difference between nose and forehead temperature [108,110]. Moreover, thermal IR imaging was demonstrated to be a valuable indicator of the driver arousal level, from alertness to drowsiness [64,111]. A recent study employed it to detect human thermal discomfort in order to develop a fully automated climate control system in the vehicles [112]. Even in the view of automated vehicles, in complex situations, they will need human

assistance for a long time. Therefore, for road safety, driver monitoring is more relevant than ever, in order to keep the driver alert and awake.

Autonomous driving relies on the understanding of objects and scenes through images. A recent study assessed a fusion system consisting of a visible and thermal sensor on object recognition from a driving dataset, and it demonstrated that thermal images significantly improve detection accuracy [113]. Miethig et al. argued that thermal IR imaging can improve the overall performance of existing autonomous detection system and reduce pedestrian detection time [114].

In conclusion, thermal IR imaging can be greatly useful also in vehicle technology; however, further testing is needed to better understand how it would improve automated vehicles and the knowledge about cognitive states in traffic safety.

## 10. Discussion

The advancement in robotics, especially in social robotics, is breathtaking. Social robots could potentially revolutionize the way of taking care of the sick or the elderly people, the way of teaching and learning, and even the definition of the concept of companionship. Nowadays, social robots hold the promise of extending life expectancies and improving health and the quality of life. In this review, the impact that social robots have in fields such as education, healthcare, and tourism is briefly surveyed. In all these areas, they are mainly intended to improve or protect the lifestyle and health, both physical and mental, of the human user. In fact, robots can also help socially impaired people relate to others, practice empathetic behaviors, and act as a steppingstone toward human contact. However, the most important and desirable requirement is that the robot meets the person's needs. Robots actually need to be able to recognize the interlocutor's affective state to communicate naturally with him/her and to engage him/her not only on a cognitive level but on an emotional level as well.

To be able to get information about the partner's affective state, there are at least two possibilities. Either the user has to explicit and voluntarily express information about his/her emotions to the robot—for example, using natural language or facial expressions and gestures, or the robot has to recognize involuntary affective information from physiological measures, such as respiration, heart rate, skin conductance, skin temperature, and blood pressure. In the present review, the use of an ecological technology is promoted, such as the thermal IR imaging-based affective computing technique. It aimed to facilitate HRI by endowing the robots with the capability of autonomously identifying the person's emotional state. Thermal IR imaging, by recording facial cutaneous temperature and its topographic distribution, is able to identify specific features clearly correlated to emotional state and measures associated with standard physiological signals of the sympathetic activity. Emotional state, in fact, determines a redistribution of the blood in the vessels through vasodilation or vasoconstriction phenomena, which are regulated by ANS. These phenomena can be identified and monitored over time because they produce a thermal variation on the skin. The thermal IR imaging technique is already validated in the literature for emotion recognition tasks.

Regarding emotion recognition, gold standard techniques such as speech and facial expression analysis were mentioned. However, it is worth mentioning that to distinguish between emotions, different models or theories have been so far developed and used by psychologists or cognitive neuroscientists. Thus, it is difficult to take a theory of one research field, such as psychology, and apply it to another, such as HRI. This remains an open discussion issue in the HRI research field. For instance, in speech analysis, the emotion recognition models developed using the utterances of a particular language usually do not yield appreciably good recognition performance for utterances from other languages [115]. On the other hand, in facial expression and gesture analysis, two main theories are currently established in emotion research: a discrete approach, claiming the existence of universal "basic emotions" [116], and a dimensional approach, assuming the existence of two or more major dimensions that can describe different emotions and distinguish between them [117]. The thermal IR imaging technique, i.e., the ROI's temperature modulation analysis, has demonstrated to be suitable for emotion recognition based on both the basic emotion approach and on the dimensional theory of

the emotion (as reported in Section 4). This makes thermal IR imaging a cross-cutting and ubiquitous technique in the area of emotion recognition and consequently a valuable contribution in HRI studies. Of course, thermal IR imaging is not the first and unique technology explored for emotion recognition through physiological measures in HRI, but it seems to be one of the most ecological. In this review, the major challenges toward the application of this technique in HRI fields has been highlighted, bridging the gap between the constrained laboratory setting and the real-world scenario. To this end, a few studies have been reported in the literature and were here analyzed. The application of thermal IR imaging-based Affective Computing in the field of HRI has been reviewed as well. Interesting results were reported by Goulart et al, in which the developed system was able to recognize emotions, achieving 85.75% accuracy [86]. To some extent, such accuracy can be comparable with gold standard techniques in emotion recognition for HRI, such as facial expression analysis and speech tone analysis.

An important aspect that further draws attention to this technique is its adaptability in applications where there is interaction between social robots and challenging populations such as neonates. This was the case of RAVE, the learning tool composed of an avatar and a social robot, which was designed to facilitate langue learning during the critical developmental period (age 6–12 months) and devolved to infants who might not otherwise receive proper language exposure. In conclusion, we believe that this review paves the way for the use of thermal IR imaging in HRI, which could endow the social robot with the capability of recognizing the interlocutor's emotions relying on involuntary physiological signals measurements. These measurements may be fed to multivariate linear [118,119] or non-linear regressors or classification algorithms [120], also relying on data-driven machine learning and deep learning approaches [121,122]. In this way, it is possible to avoid the artifact of social masking and make HRI suitable also for people who lack the ability to express emotions. Beyond robots, intelligent systems such as autonomous vehicles or even smart buildings could also benefit from this technique.

## 11. Conclusions

HRI is a relatively young discipline that has attracted a lot of attention in recent years due to the increasing availability of complex robots and people's exposure to such robots in their daily lives. Moreover, robots are increasingly being developed for real-world application areas, such as education, healthcare, eldercare, and other assistive applications. A natural HRI is crucial for the beneficial influence that robots can have in human life. Understanding the interlocutor's need and affective state during the interplay is the foundation of a human-like interaction. To this end, an ecological technology such as thermal IR imaging, which can provide information about physiological parameters associated to the subject affective state, was here presented and surveyed. The technology can provide the ground for the further development of robust social robots and to facilitate HRI. Thermal IR imaging has already been validated in the literature in the fields of emotion recognition. This review can act as a guideline to, and foster, the use of thermal IR imaging-based affective computing in HRI applications, which is intended to support a natural HRI, with special regard to those who find difficult to express emotions.

**Author Contributions:** Conceptualization, C.F., A.M.; investigation, C.F., D.C., A.M.C., D.P.; writing—original draft preparation, C.F.; writing—review and editing D.C., A.M.C., D.P.; supervision, A.M.; funding acquisition, A.M. All authors have read and agreed to the published version of the manuscript.

**Funding:** This research was funded by PON FESR MIUR R&I 2014-2020 - Asse II - ADAS+, ARS01_00459 and PON MIUR SI-ROBOTICS, ARS01_01120.

**Conflicts of Interest:** The authors declare no conflict of interest.

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
