# Peer review of "Thermal Infrared Imaging-Based Affective Computing and Its Application to Facilitate Human Robot Interaction: A Review"

_applsci, doi:10.3390/app10082924_

Round 1

Reviewer 1 Report

The paper is a review of state-of-the-art in combining infrared imaging, recognition of human affect and human-robot interaction.

The paper is interesting, however in my opinion study was not well designed from methodological perspective.

There are methods for performing systematic literature reviews and methods for reporting them - please take look at PRISMA or works by Kitchenham.

This report is missing the descriptions of:

  • paper inclusion criteria
  • keywords searched
  • databases browsed
  • paper selection process and elimination (numbers) 

Some of the references are missing eg. line 357 mentions works by Ruminski et al. but no reference to such paper is provided in the reference section.

Author Response

The paper is a review of state-of-the-art in combining infrared imaging, recognition of human affect and human-robot interaction.

The paper is interesting, however in my opinion study was not well designed from methodological perspective.

There are methods for performing systematic literature reviews and methods for reporting them - please take look at PRISMA or works by Kitchenham.

This report is missing the descriptions of:

  • paper inclusion criteria
  • keywords searched
  • databases browsed
  • paper selection process and elimination (numbers)

The authors thank the reviewer for the relevant indication. According to the reviewer’s suggestion and following the Kitchenham guidelines, the title has been changed and a detailed explanation of the search methods adopted has been added.

The new title is the following:

“Thermal infrared imaging-based affective computing and its application to facilitate human robot interaction: a review”

The added paragraph (page 3 and 4) is the following:

1.1 Study organization and search processing method

This study has been carried out as a systematic literature review based on the original guidelines as proposed by Kitchenham [36]. The work aims to highlight the positive impact that the thermal IR imaging technology can have on HRI application, facilitating and enhancing this interaction by recognizing the human affective state. For this purpose, the research questions (RQ) addressed by this study were:

RQ1. What is the broad social impact of a facilitated HRI? Or Why facilitate HRI is important?

RQ2. What are the scientific bases for thermal IR imaging as affective state recognition technique?  

RQ3. What are the limitations of current research?

RQ4. Has thermal IR imaging already been addressed in the HRI field?

One of the goals of this study was to make this review as inclusive and practical as possible, therefore, the databases searched were both Scopus and Google Scholar. All papers published in conferences and journals between 2000 and 2020 were considered. Papers published from the year 2000 were considered since the word “Affective computing” was coined by Picard that year [6] and “Affective computing” started to be applied in the HRI field. For each research question, a search process was applied.

Concerning RQ1 the search was based on the words “facilitate” and “Human-robot Interaction”. Review papers and all papers that did not refer to a user study or that did not relate to HRI were excluded, which reduced the considered pool to 155 papers. Final exclusion was related to all papers that did not report experimental applications of human interaction. 18 papers were discussed in the review concerning these keywords.

With respect to RQ2 and RQ3, the searched keywords were: "thermal imaging" OR "IR imaging" OR "thermography" AND "emotion recognition" OR "Affective Computing" OR "emotion". The results were scanned to identify whether the considered works reported on thermal IR imaging for human emotion recognition; papers not related to it were excluded.  The resulting papers were analyzed, grouped based on their experimental applications, strengths and limitations indicated, and 46 were reported in the present work. 

Finally, as for the RQ4, keywords "thermal imaging" OR "IR imaging" OR "thermography" AND "Human-Robot interaction", were searched. Exclusion criteria regarded all the results that were not conference or journal papers actually related to thermal IR imaging applied in HRI field. From the latter, 20 papers of thermal IR imaging-based affective computing were selected and included in this work.”

 [36] Kitchenham, B. Procedures for Performing Systematic Reviews. 33.

Some of the references are missing eg. line 357 mentions works by Ruminski et al. but no reference to such paper is provided in the reference section.

The authors thank the reviewer for finding the mistakes, that were corrected.

  1. Movellan, J.; Eckhardt, M.; Virnes, M.; Rodriguez, A. Sociable robot improves toddler vocabulary skills. In Proceedings of the Proceedings of the 4th ACM/IEEE international conference on Human robot interaction; 2009; pp. 307–308.
  2. Banks, M.R.; Willoughby, L.M.; Banks, W.A. Animal-assisted therapy and loneliness in nursing homes: use of robotic versus living dogs. J. Am. Med. Dir. Assoc. 2008, 9, 173–177.
  3. Ruminski, J.; Kwasniewska, A. Evaluation of respiration rate using thermal imaging in mobile conditions. In Application of Infrared to Biomedical Sciences; Springer, 2017; pp. 311–346.

Reviewer 2 Report

The paper presented is essentially a state-of-the-art review on how to measure affects and emotions using infrared technology.

The first introductory section presents the context of this review, i.e. the use of these technologies for human-robot interaction. The authors position their credo by pointing out that in order to facilitate HRI it is necessary to have a good understanding of human emotional reactions. They compare the advantages/disadvantages of a characterization made by classical cameras or using infrared techniques. The second paragraph on page 2 of the introduction is very long and makes a lot of claims without any scientific references. The authors must sustain their claims by references.

Section 2 emphasizes the need to make human-robot interaction more natural and efficient. The authors cite and describe several areas of use of social robots, namely the fields of education, health, and tourism, in order to highlight the potential benefits and positive impacts of using robots in these fields.

Section 3 goes into more detail, describing the measurements made by thermal infrared imaging on the subjects' faces, in order to characterize their emotions. Table 1 has a double “Nose” entry. Please check it. In paragraph starting at line 254, it is strange that the authors did not mention the PAD model from Mehrabian (1996) as it is widely used, even in HRI. Last paragraph of section 3 insists on specific condition to use thermal imaging. The authors might put an emphasis on it, maybe putting a subsection with a clear title before it.

Section 4 then insists on the use of mobile technologies, which is understandable, if one wishes to embark in a robot these infrared measurement means.

Section 5, more surprisingly, describes the methods used to combine infrared technologies and affective computing, “outside laboratory”, while section 6 focuses on this same combination but more specifically for HRI. It is not clear why the authors have divided this part of their review into two sections.

In section 7, the authors discuss how to use different dimensions to classify emotions (with basics emotions), and the advantages of using infrared technologies to measure them. However, we regret that there is no comparison with other means of determining emotions, such as the use of speech signal, which is a vector (which is well known) very often used by humans to express their emotions.

We also regret that the authors did not consider extending their review to other types of complex intelligent systems that are not humanoid robots, even if their intelligent behaviors would also allow them to be classified as "robots", such as autonomous vehicles or smart buildings for instance.

Section 8 is a short conclusion

Throughout these sections the authors cite and compare a large number of literature works: 90 references are listed in the bibliography section. These references are almost adequate and a large part is recent.

Our main comment concerns the way sections 1 and 2 are written: the authors are far too affirmative about the “desired” anthropomorphism of robots. We give only two examples here, but this criticism is somewhat echoed throughout both sections.  Lines 39 and 40 the authors write: "...robots... ...must be able to socially interact." We find the "must be" too affirmative because for us, it depends a lot on the application.  Similarly, the second sentence of section 2 "The acceptance... ... and tourism" seems to us to be false... or at least it seems inconsistent, even dangerous, to follow such a statement!  Perhaps this is due to the fact that the authors cite works that are no longer really recent to support their assertions: in particular, references 1 to 5 are works carried out in the years 2001 to 2008.

Indeed, some recent works criticize the anthropomorphism of social robots. We will not quote them all in this commentary, but only those that come to mind.

For example, Sherry Turkle regrets a loss of authenticity and fears that seductive relationships with robots (assuming they are less difficult than relationships with humans) may lead people to avoid interacting with friends and family (2010).

Privacy is one of the main concerns about the manipulation of user behavior by technology.  Indeed, the emotional engagement inherent in the use of social robots may induce people to exchange personal information for functional rewards. Another danger is that people may reveal more information about themselves than they would voluntarily and knowingly enter into a database (Thomasen 2016).

We complete our remarks by summarizing some of the findings of Kate Darling (2015, 2016) which we quote here almost verbatim: The question of whether robots can actually change people's long-term behaviour, either positively or negatively, remains unanswered. For this reason, it seems much more prudent to distinguish between cases where the use of anthropomorphism can be encouraged, and cases where it is better not to.  For robots that are not designed in an intrinsically social way, we should consider discouraging the use of anthropomorphism: instead of considering "personification" and anthropomorphisation as "harmless fun", those who implement robotic technology should be more aware of collateral effects.

Finaly, we can also cite the work of B.R. Duffy (2003) who summarizes the debate (already in 2003) in his conclusion and of which we quote some lines: “Experiments comparing a poor implementation of ahuman-like software character to a better implementation of a dog-like character promotes the idea that a restrained degree of anthropomorphic form and function is the optimal solution for an entity that is not a human”… and asks the question a few lines later: “Should  robots  become  the synthetic human?”  

That is why we suggest that the authors be a little more "neutral" in the wording of some of their statements concerning the anthorpomorphism. The anthropomorphization of systems now raises real debates, not only in robotics but also in artificial intelligence.

Our last remark concerns the title of the study. The authors could add the word "Survey" in the title, as their work is a (well done) survey. They should put more emphasis on "thermal infrared imaging", rather than HRI, as the authors do not bring any work or personal results on the "application to HRI" part. We would propose “Survey on thermal infrared imaging-based affective computing and its application to facilitate human robot interaction” (or something similar). To us, it will increase also the impact article.

A general form comment. The authors should avoid sentence lie “The authors of [63] …” (an example from line 470, but there are many other examples). Usually, sentences should be readable with citation numbers.

Cited references:

Mehrabian, A (1996). Pleasure-arousal-dominance: A general framework for describing and measuring individual differences in Temperament. Current Psychology 14, 261–292 (1996). https://doi.org/10.1007/BF02686918

Darling, Kate (2015), 'Who's Johnny?' Anthropomorphic Framing in Human-Robot Interaction, Integration, and Policy (March 23, 2015). ROBOT ETHICS 2.0, eds. P. Lin, G. Bekey, K. Abney, R. Jenkins, Oxford University Press, 2017, Forthcoming. Available at SSRN: https://ssrn.com/abstract=2588669 or http://dx.doi.org/10.2139/ssrn.2588669

Darling, Kate (2016). “Extending Legal Protections to Social Robots. The Effects of Anthropomorphism, Empathy, and Violent Behavior towards Robotic Objects.” In Robot Law, edited by Ryan Calo, Michael Froomkin, and Ian Kerr, 213-34. Edward Elgar.

Turkle, Sherry (2010). “In good company? On the threshold of robotic companions.” In Close Engagements with Artificial Companions: Key Social, Psychological, Ethical and Design Issues, 3-10. John Benjamins Publishing Company.

Thomasen, Kristen (2016). “Examining the Constitutionality of Robot Enhanced Interrogation.” In Robot Law, edited by Ryan Calo, Michael Froomkin, and Ian Kerr, 306-332. Edward Elgar.

Brian R. Duffy (2003) “Anthropomorphism and the social robot” Robotics and Autonomous Systems, 42, 2003, 177-190.

Author Response

The paper presented is essentially a state-of-the-art review on how to measure affects and emotions using infrared technology.

The first introductory section presents the context of this review, i.e. the use of these technologies for human-robot interaction. The authors position their credo by pointing out that in order to facilitate HRI it is necessary to have a good understanding of human emotional reactions. They compare the advantages/disadvantages of a characterization made by classical cameras or using infrared techniques. The second paragraph on page 2 of the introduction is very long and makes a lot of claims without any scientific references. The authors must sustain their claims by references.

The authors agree with the reviewer. The following references have been added to the second paragraph of page 2 to support their claims.

  1. Russell, J.A.; Bachorowski, J.-A.; Fernández-Dols, J.-M. Facial and vocal expressions of emotion. Annu. Rev. Psychol. 2003, 54, 329–349.
  2. Fernandes, A.; Helawar, R.; Lokesh, R.; Tari, T.; Shahapurkar, A.V. Determination of stress using blood pressure and galvanic skin response. In Proceedings of the 2014 International Conference on Communication and Network Technologies; IEEE, 2014; pp. 165–168.
  3. Bradley, M.M.; Lang, P.J. Measuring emotion: Behavior, feeling, and physiology. Cogn. Neurosci. Emot. 2000, 25, 49–59.
  4. Schachter, S.; Singer, J. Cognitive, social, and physiological determinants of emotional state. Psychol. Rev. 1962, 69, 379.
  5. Knapp, R.B.; Kim, J.; André, E. Physiological signals and their use in augmenting emotion recognition for human–machine interaction. In Emotion-oriented systems; Springer, 2011; pp. 133–159.
  6. Reynolds, C.; Picard, R. Affective sensors, privacy, and ethical contracts. In Proceedings of the CHI’04 extended abstracts on Human factors in computing systems; 2004; pp. 1103–1106.
  7. Sebe, N.; Sun, Y.; Bakker, E.; Lew, M.S.; Cohen, I.; Huang, T.S. Towards authentic emotion recognition. In Proceedings of the 2004 IEEE International Conference on Systems, Man and Cybernetics (IEEE Cat. No. 04CH37583); IEEE, 2004; Vol. 1, pp. 623–628.
  8. Schmidt, K.L.; Cohn, J.F. Human facial expressions as adaptations: Evolutionary questions in facial expression research. Am. J. Phys. Anthropol. Off. Publ. Am. Assoc. Phys. Anthropol. 2001, 116, 3–24.
  9. Crivelli, C.; Fridlund, A.J. Facial displays are tools for social influence. Trends Cogn. Sci. 2018, 22, 388–399.
  10. Wioleta, S. Using physiological signals for emotion recognition. In Proceedings of the 2013 6th International Conference on Human System Interactions (HSI); IEEE, 2013; pp. 556–561.
  11. Jerritta, S.; Murugappan, M.; Nagarajan, R.; Wan, K. Physiological signals based human emotion recognition: a review. In Proceedings of the 2011 IEEE 7th International Colloquium on Signal Processing and its Applications; IEEE, 2011; pp. 410–415.
  12. Cross, C.B.; Skipper, J.A.; Petkie, D.T. Thermal imaging to detect physiological indicators of stress in humans. In Proceedings of the Thermosense: Thermal Infrared Applications XXXV; International Society for Optics and Photonics, 2013; Vol. 8705, p. 87050I.

Section 2 emphasizes the need to make human-robot interaction more natural and efficient. The authors cite and describe several areas of use of social robots, namely the fields of education, health, and tourism, in order to highlight the potential benefits and positive impacts of using robots in these fields.

Section 3 goes into more detail, describing the measurements made by thermal infrared imaging on the subjects' faces, in order to characterize their emotions. Table 1 has a double “Nose” entry. Please check it.

The table has been corrected, please refer to the new Table 1 page 7.

In paragraph starting at line 254, it is strange that the authors did not mention the PAD model from Mehrabian (1996) as it is widely used, even in HRI.

The authors want to thank the reviewer for the relevant suggestion; the work has been now discussed in the text. The paragraph (page 8) has been modified as follows:

“Beyond the basic emotions, thermal IR imaging has been used to characterize the two dimensions of emotions, such as valence (pleasant vs. unpleasant) and arousal (low vs. high). Emotions dimensions are a crucial aspect in the affective research field, on which most of the studies on emotions recognition are based on. The most commonly used models representing emotions dimension in HRI are the Pleasure, Arousal, Dominance (PAD) emotional state model [58] and the circumplex model of affect [59]. The PA dimensions of PAD were developed into the circumplex model, which indeed assume that any emotion might be described with two continuous dimensions of valence and arousal [60]. Valence dimension indicates whether the subject's current emotional state is positive or negative. Arousal, on the other hand, indicates whether the subject is responsive or not, at that given moment and for that given stimulus, and how active he/she is. In particular, the theory of the dimension of the emotions proposes that the emotional states are not discrete categories but rather a result of varying degrees of their dimensions. A graphical representation of circumplex model of affect developed by Russel is reported in Figure 3. Joy, for example, is characterized as the product of strong activation in the neural systems associated with positive valence or pleasure together with moderate activation in the neural systems associated with arousal (i.e. low arousal). Emotions other than joy likewise arise from the same two-dimensional systems but differ in the degree or extent of activation. This allows to characterize also complex emotions such as love or happiness other than the basic ones. The analysis of the emotion recognition solutions reveals that there is no one commonly accepted standard model for emotion representation. Dimensional adaptation of Ekman’s six basic emotions and the Circumplex or PAD model are the ones widely adopted in emotion recognition solutions [61].”

  1. Mehrabian, A. Pleasure-arousal-dominance: A general framework for describing and measuring individual differences in temperament. Curr. Psychol. 1996, 14, 261–292

Last paragraph of section 3 insists on specific condition to use thermal imaging. The authors might put an emphasis on it, maybe putting a subsection with a clear title before it.

The following subsection (page 10) has been added:

"3.1. Limits of current thermal IR imaging for HRI applications

Studies reviewed in this section revealed thermal IR imaging ability to monitor physiological signs and affective states…”

Section 4 then insists on the use of mobile technologies, which is understandable, if one wishes to embark in a robot these infrared measurement means.

Section 5, more surprisingly, describes the methods used to combine infrared technologies and affective computing, “outside laboratory”, while section 6 focuses on this same combination but more specifically for HRI. It is not clear why the authors have divided this part of their review into two sections.

The authors have better described the rationale of separating the topics in two paragraphs. A detailed explanation has been reported at the end of section 3.1 (page 11), which includes the importance of having section 5 as a separate section.

“In conclusion, by analyzing the studies reported in this section, two main constraints were identified that are not suitable for HRI applications. Those are: i) the use of high end and sized thermal imaging systems and ii) the circumstances in which the studies were conducted i.e. in restricted laboratory settings. On the other hand, HRI applications require daily life scenarios, eventually suitable for outdoor use, and technology embedded in commercial social robots, i.e. low-cost miniaturized sensors. In sections 4 and 5 those limits are addressed, in order to highlight the new improvements developed in recent studies. A special emphasis was placed on these two sections, as they deal with crucial aspects for the use of thermal IR imaging in the HRI field. Finally, the last section focuses on the actual state of the art of thermal IR imaging-based affective computing applications.”

In section 7, the authors discuss how to use different dimensions to classify emotions (with basics emotions), and the advantages of using infrared technologies to measure them. However, we regret that there is no comparison with other means of determining emotions, such as the use of speech signal, which is a vector (which is well known) very often used by humans to express their emotions.

Speech analysis for emotion recognition, which were already briefly mentioned throughout the manuscript, has been further commented in section 7 (page 17), as follows:

“Regarding the emotion recognition, gold standard techniques such as speech and facial expression analysis were mentioned. However, it is worth mentioning that to distinguish between emotions, different models or theories, have been, so far, developed and used by psychologists or cognitive neuroscientists. Thus, it is difficult to take a theory of one research field, like psychology, and apply it to another, like HRI. This remain an open discussion issue in the HRI research field. For instance, in speech analysis the emotion recognition models developed using the utterances of a particular language usually don’t yield appreciably good recognition performance for utterances from other languages [116].  On the other hand, in facial expression and gesture analysis, two main theories are currently established in emotion research: a discrete approach, claiming the existence of universal “basic emotions” [117], and a dimensional approach, assuming the existence of two or more major dimensions which can describe different emotions and distinguish between them [118]. Thermal IR imaging technique, i.e. ROI’s temperature modulation analysis, has demonstrated to be suitable for emotion recognition based on both the basic emotion approach and on the dimensional theory of the emotion (as reported in section 3). This makes thermal IR imaging a cross-cutting and ubiquitous technique in the area of emotion recognition, and consequently a valuable contribution in HRI studies. Of course, thermal IR imaging is not the first and unique technology explored for emotion recognition through physiological measure in HRI, but it seems to be one of the most ecological. In this review, the major challenges toward the application of this technique in HRI fields has been highlighted, bridging the gap between the constrained laboratory setting and the real-world scenario. To this end, few studies have been reported in literature and were here analyzed. Application of thermal IR imaging-based affective computing in the field of HRI have been reviewed as well. Interesting results were reported by Goulart et al, in which the developed system was able to recognize emotions, achieving 85.75% accuracy [87]. To some extent, such accuracy can be comparable with gold standard techniques in emotion recognition for HRI, such as facial expression analysis and speech tone analysis.”

We also regret that the authors did not consider extending their review to other types of complex intelligent systems that are not humanoid robots, even if their intelligent behaviors would also allow them to be classified as "robots", such as autonomous vehicles or smart buildings for instance.

The authors thank for the suggestion; the authors added a section (page 16) to include literary review on autonomous vehicle and drive-assistance system. Unfortunately, the authors could not find many works on thermal imaging and smart building.

“6.1. Thermal IR imaging-based affective computing in intelligent systems such as driver-assistance systems or autonomous vehicles.

Besides humanoid robots, systems such as driver-assistance systems or autonomous vehicles can also be classified as “robot” due to their intelligent behaviors. Indeed, driving functions are becoming increasingly automated, consequently, motorists could run the risk of being cognitively removed from the driving process. Thermal IR imaging was demonstrated to be a valid technique for assessing variation in cognitive load [52,108,109]. Most of all, it could enable sensing the real-time state of motorists non-invasively i.e. without disrupting driving-related tasks and, unlike RGB cameras, independently from external light conditions [110]. Studies of thermal IR imaging in driving contexts stated that a rise in mental workload leads to an increase in the difference between nose and forehead temperature [109,111]. Moreover, thermal IR imaging was demonstrated to be a valuable indicator of the driver arousal level, from alertness to drowsiness [64,112]. A recent study employed it to detect human thermal discomfort in order to develop a fully automated climate control system in the vehicles [113]. Even in view of automated vehicles, in complex situations, they will need human assistance for a long time. Therefore, for road safety, driver monitoring is more relevant than ever, in order to keep the driver alert and awake.

Autonomous driving relies on the understanding of objects and scenes through images. A recent study assessed a fusion system consisting of a visible and thermal sensor on object recognition, from driving dataset, and demonstrated that thermal images significantly improve detection accuracy [114]. Miethig et al. argued that thermal IR imaging can improve the overall performance of existing autonomous detection system and reduce pedestrian detection time [115].

In conclusion, thermal IR imaging can be greatly useful also in vehicle technology, however further testing is needed to better understand how it would improve automated vehicles, and the knowledge about cognitive states in traffic safety.”

 In addition, smart building and autonomous vehicles have been cited in this section (page18), as follows:

“In this way, it is possible to avoid the artifact of social masking and make HRI suitable also for people who lack the ability to express emotions. Beyond robot, intelligent systems such as autonomous vehicles or even smart buildings could also benefit from this technique “.

Section 8 is a short conclusion

Throughout these sections the authors cite and compare a large number of literature works: 90 references are listed in the bibliography section. These references are almost adequate and a large part is recent.

Our main comment concerns the way sections 1 and 2 are written: the authors are far too affirmative about the “desired” anthropomorphism of robots. We give only two examples here, but this criticism is somewhat echoed throughout both sections.  Lines 39 and 40 the authors write: "...robots... ...must be able to socially interact." We find the "must be" too affirmative because for us, it depends a lot on the application.  Similarly, the second sentence of section 2 "The acceptance... ... and tourism" seems to us to be false... or at least it seems inconsistent, even dangerous, to follow such a statement!  Perhaps this is due to the fact that the authors cite works that are no longer really recent to support their assertions: in particular, references 1 to 5 are works carried out in the years 2001 to 2008.

Indeed, some recent works criticize the anthropomorphism of social robots. We will not quote them all in this commentary, but only those that come to mind.

For example, Sherry Turkle regrets a loss of authenticity and fears that seductive relationships with robots (assuming they are less difficult than relationships with humans) may lead people to avoid interacting with friends and family (2010).

Privacy is one of the main concerns about the manipulation of user behavior by technology.  Indeed, the emotional engagement inherent in the use of social robots may induce people to exchange personal information for functional rewards. Another danger is that people may reveal more information about themselves than they would voluntarily and knowingly enter into a database (Thomasen 2016).

We complete our remarks by summarizing some of the findings of Kate Darling (2015, 2016) which we quote here almost verbatim: The question of whether robots can actually change people's long-term behaviour, either positively or negatively, remains unanswered. For this reason, it seems much more prudent to distinguish between cases where the use of anthropomorphism can be encouraged, and cases where it is better not to.  For robots that are not designed in an intrinsically social way, we should consider discouraging the use of anthropomorphism: instead of considering "personification" and anthropomorphisation as "harmless fun", those who implement robotic technology should be more aware of collateral effects.

Finaly, we can also cite the work of B.R. Duffy (2003) who summarizes the debate (already in 2003) in his conclusion and of which we quote some lines: “Experiments comparing a poor implementation of ahuman-like software character to a better implementation of a dog-like character promotes the idea that a restrained degree of anthropomorphic form and function is the optimal solution for an entity that is not a human”… and asks the question a few lines later: “Should  robots  become  the synthetic human?” 

That is why we suggest that the authors be a little more "neutral" in the wording of some of their statements concerning the anthorpomorphism. The anthropomorphization of systems now raises real debates, not only in robotics but also in artificial intelligence.

Sections 1 and 2 have been reframed according to the reviewer’s advices. A paragraph has been added in section 1 (page 2) to better discuss the reviewer’s suggestions, and to place more emphasis on the work’s purpose i.e. to facilitate HRI, rather than robot anthropomorphizing.

Section 1 has been modified as follow:

“This goes especially with children, who “are unlikely to only use a robot as a tool, and they will undoubtedly have some sort of interaction that can be considered social” [3]. Therefore, concerning robots designed to interact with humans and, in particular, with children, it would be desirable for them to be able to socially interact. To facilitate the interaction, robots should be easy to use, and they should be built to understand and correctly respond to different needs. As robots gain utility, and thereby influence on society, research in HRI is becoming increasingly important. HRI primarily deals with social robotics and includes relevant aspects of robot action, perception, and cognition. The development of social robots focuses on the design of living machines that humans would perceive as realistic, effective, communicative and cooperative [4]. To this purpose, social robots may be able to express, through their shapes and behaviors, a certain degree of “intelligence” [5]. This skill entails the whole set of social and perceptual abilities of the robot, delivering a human-like interaction.

However, on the other hand, the matter about the anthropomorphism of social robots is still strongly debated. Indeed, the inclination to anthropomorphize the existing technology could have negative consequences [6] and even lead to an unrecognized change in human relationships [7] or invasion of privacy [8]. It would be therefore prudent to distinguish cases in which the use of anthropomorphism can be encouraged and cases in which it would be better not to [9]. Darling [10] argues that anthropomorphic framing is desirable where it enhances the function of the technology. Especially, it should be encouraged for social robots and discouraged for robots that are not designed in an intrinsically social way [9,10]. The present review is focused on social robots and in this context, their anthropomorphism is aimed at facilitating the interaction with humans.”

Section 2 (page 4) has been reframed as follow:

“Robotic technologies and HRI are being increasingly integrated in real-life contexts. Modern society creates ever more spaces where robot technology is purposed to interact with people. Robots applications can range from education to communication, assistance, entertainment, healthcare, and tourism. Hence the need to better understand how robotic technologies shape the social contexts in which they are used [37].”

References

  1. Duffy, B.R. Anthropomorphism and the social robot. Robot. Auton. Syst. 2003, 42, 177–190.
  2. Turkle, S. In good company?: On the threshold of robotic companions. In Close engagements with artificial companions; John Benjamins, 2010; pp. 3–10.
  3. Thomasen, K. Examining the constitutionality of robot-enhanced interrogation. In Robot Law; Edward Elgar Publishing, 2016.
  4. Darling, K. Extending legal protection to social robots: The effects of anthropomorphism, empathy, and violent behavior towards robotic objects. In Robot law; Edward Elgar Publishing, 2016.
  5. Darling, K. ’Who’s Johnny?’Anthropomorphic Framing in Human-Robot Interaction, Integration, and Policy. Anthr. Fram. Hum.-Robot Interact. Integr. Policy March 23 2015 ROBOT ETHICS 2015, 2.

Our last remark concerns the title of the study. The authors could add the word "Survey" in the title, as their work is a (well done) survey. They should put more emphasis on "thermal infrared imaging", rather than HRI, as the authors do not bring any work or personal results on the "application to HRI" part. We would propose “Survey on thermal infrared imaging-based affective computing and its application to facilitate human robot interaction” (or something similar). To us, it will increase also the impact article.

The titled has been modified according to the reviewer’s suggestion.

“Thermal infrared imaging-based affective computing and its application to facilitate human robot interaction: a review”

A general form comment. The authors should avoid sentence lie “The authors of [63] …” (an example from line 470, but there are many other examples). Usually, sentences should be readable with citation numbers.

Those sentences have been changed throughout the manuscript as suggested by the reviewer.

Cited references:

Mehrabian, A (1996). Pleasure-arousal-dominance: A general framework for describing and measuring individual differences in Temperament. Current Psychology 14, 261–292 (1996). https://doi.org/10.1007/BF02686918

Darling, Kate (2015), 'Who's Johnny?' Anthropomorphic Framing in Human-Robot Interaction, Integration, and Policy (March 23, 2015). ROBOT ETHICS 2.0, eds. P. Lin, G. Bekey, K. Abney, R. Jenkins, Oxford University Press, 2017, Forthcoming. Available at SSRN: https://ssrn.com/abstract=2588669 or http://dx.doi.org/10.2139/ssrn.2588669

Darling, Kate (2016). “Extending Legal Protections to Social Robots. The Effects of Anthropomorphism, Empathy, and Violent Behavior towards Robotic Objects.” In Robot Law, edited by Ryan Calo, Michael Froomkin, and Ian Kerr, 213-34. Edward Elgar.

Turkle, Sherry (2010). “In good company? On the threshold of robotic companions.” In Close Engagements with Artificial Companions: Key Social, Psychological, Ethical and Design Issues, 3-10. John Benjamins Publishing Company.

Thomasen, Kristen (2016). “Examining the Constitutionality of Robot Enhanced Interrogation.” In Robot Law, edited by Ryan Calo, Michael Froomkin, and Ian Kerr, 306-332. Edward Elgar.

Brian R. Duffy (2003) “Anthropomorphism and the social robot” Robotics and Autonomous Systems, 42, 2003, 177-190.

The authors thank for the suggestion; The suggested references were added to the manuscript when needed.

Reviewer 3 Report

The authors aim to review thermal infrared imaging-based affective computing used in social robotics.

In general, the paper is well written and the content has a certain contribution to the field.

I would recommend acceptance with some required corrections.

  1. Serious mistake: the robot is NAO, not Pepper. 
  2. The authors should go to original sources of information as much as possible. For example, most of the information about using social robots in autism therapy comes from ref [23] which seems to be from a predator journal. I suggest the author remove this citation and go for better ones.

Author Response

The authors aim to review thermal infrared imaging-based affective computing used in social robotics.

In general, the paper is well written and the content has a certain contribution to the field.

The authors thank the reviewer for the positive feedback.

I would recommend acceptance with some required corrections.

  1. Serious mistake: the robot is NAO, not Pepper.

Thank you for finding the mistake. The image was modified.

  1. The authors should go to original sources of information as much as possible. For example, most of the information about using social robots in autism therapy comes from ref [23] which seems to be from a predator journal. I suggest the author remove this citation and go for better ones.

Reference 23 has been changed with the following references:

  1. Hodges, H.; Fealko, C.; Soares, N. Autism spectrum disorder: definition, epidemiology, causes, and clinical evaluation. Transl. Pediatr. 2020, 9, S55.
  2. Cabibihan, J.-J.; Javed, H.; Ang, M.; Aljunied, S.M. Why robots? A survey on the roles and benefits of social robots in the therapy of children with autism. Int. J. Soc. Robot. 2013, 5, 593–618.
  3. Sartorato, F.; Przybylowski, L.; Sarko, D.K. Improving therapeutic outcomes in autism spectrum disorders: Enhancing social communication and sensory processing through the use of interactive robots. J. Psychiatr. Res. 2017, 90, 1–11.

Round 2

Reviewer 1 Report

The paper improved significantly in reporting SLR in methodological way.

However, some details are missing, issues to address:

  • provide the number of papers found in Scopus and in Google Scholar separately and the overlapping papers count;
  • please provide info on the fields you were searching in each database (title, abstract, keywords, other?);
  • please provide info, whether reported number of papers were after the basic search or after the manual review;
  • what was the procedure of manual review - titles OR abstract or whole papers were scanned?
  • where the subsets for research questions treated jointly? what was the overlap between them?

Minor issues:

1.
line 13 "robots are becoming a substantial part of the modern society" - it's a bit exaggerated expression, you might replace "robots" with "food", "computers" or anything, moreover, its not supported with the paper, please consider re-wording

2.
line 24:
"It is a validated technology that allows the
non-obtrusive monitoring of physiological parameters and understanding of affective needs. " - I can agree with the first statement, but not the second one - the technology mentioned DOES NOT allow to understand affective needs in any meaningful way, consider re-writing

3.
Sentence lines 40-41 - unclear, check sentence syntax or grammar? 

4.
line 46 - to this purpose --> for this purpose

5.
line 85-86
", they may deliver much more reliable data about the emotional processes than visual channels. " - they deliver data on emotional SYMPTOMS, not processes 

6.
There is no need to distinguish section 1.1 if there is no section 1.2

7.
Overtime --> over time
(2 or 3 times throughout the paper, both are grammatically correct, but 'overtime' is extra hours (a noun) and 'over time' is an adverb expression meaning 'gradually in time', perhaps that what you meant)

Author Response

The paper improved significantly in reporting SLR in methodological way.

However, some details are missing, issues to address:

provide the number of papers found in Scopus and in Google Scholar separately and the overlapping papers count;

please provide info on the fields you were searching in each database (title, abstract, keywords, other?);

please provide info, whether reported number of papers were after the basic search or after the manual review;

what was the procedure of manual review - titles OR abstract or whole papers were scanned?

where the subsets for research questions treated jointly? what was the overlap between them?

The authors thank the reviewer for the relevant indication. The required information has been added to the text. The paragraph (page 4) has been modified as follows:

“One of the goals of this study was to make this review as inclusive and practical as possible, therefore, the databases searched were both Scopus and Google Scholar. All papers published in conferences and journals between 2000 and 2020 were considered. Papers published from the year 2000 were considered since the word “Affective computing” was coined by Picard that year [6] and “Affective computing” started to be applied in the HRI field. For each research question, a search process was applied.

Concerning RQ1 the search was based on the words “facilitate” and “Human-robot Interaction”. In the Scopus database, the survey was set up by searching for those words within the following fields: article title, abstract, and keywords. The basic search generated 441 results. In the Scholar database, on the other hand, the advanced search can be performed either by searching: i) the entire text or ii) the title only. Therefore, the search was based on “facilitate human-robot interaction” within the entire text. 360 results were obtained from the Scholar survey.  Review papers and all papers that did not refer to a user study or that did not relate to HRI were excluded, which reduced the considered pool to 155 papers. Within those papers, 130 were related to Scopus research and 105 from Scholar with an overlap of 80 papers. The manual review process was adopted for the final exclusion; the papers’ abstracts were scanned, and all papers that did not report experimental applications of human interaction were discarded. 18 papers were discussed in the review concerning these keywords. Among those papers, 13 papers resulted from both Scopus and Scholar research, 5 from Scholar only.

With respect to RQ2 and RQ3, the searched keywords were: "thermal imaging" OR "IR imaging" OR "thermography" AND "emotion recognition" OR "Affective Computing" OR "emotion". In Scopus database those keywords were surveyed in fields such as: article title, abstract, and keywords. Whereas in Scholar the advanced survey was carried out by searching for "thermal imaging" OR "IR imaging" OR "thermography" with at least one of these words: "Affective Computing" OR "emotion", the field searched was the entire text. The search generated 115 results in Scopus and 163 in Scholar with an overlap of 95 papers. The results were scanned trough a manual review procedure focused on the papers’ abstracts, aimed to identify whether the considered works reported on thermal IR imaging for human emotion recognition. Papers not related to it were excluded.  The resulting papers were analyzed, grouped based on their experimental applications, strengths and limitations indicated, and 46 were reported in the present work. Among those papers, 40 papers resulted from both Scopus and Scholar research, 6 from Scholar only.

 Finally, as for the RQ4, keywords "thermal imaging" OR "IR imaging" OR "thermography" AND "Human-Robot interaction", were searched. The fields checked and the procedure performed were the same reported in the previous RQs. The basic search generated 13 results in Scopus and 388 in Scholar. 10 papers resulted from Scopus search were found also in Scholar research outcome. Exclusion criteria regarded all the results that were not conference or journal papers actually related to thermal IR imaging applied in HRI field. From the latter, 20 papers of thermal IR imaging-based affective computing were selected and included in this work. Among those papers a subset of 2 papers was linked to both RQ2/RQ3 and RQ4, therefore reported in both sections.”

Minor issues:

  1.  

line 13 "robots are becoming a substantial part of the modern society" - it's a bit exaggerated expression, you might replace "robots" with "food", "computers" or anything, moreover, its not supported with the paper, please consider re-wording

The sentence in line 13 has been modified as follows:

“Over recent years, robots are increasingly being employed in several aspects of modern society. Among others, social robots have the potential to benefit education, healthcare and tourism.”

  1.  

line 24:

"It is a validated technology that allows the

non-obtrusive monitoring of physiological parameters and understanding of affective needs. " - I can agree with the first statement, but not the second one - the technology mentioned DOES NOT allow to understand affective needs in any meaningful way, consider re-writing

The sentence in line 24 has been modified as follows:

“ It is a validated technology that allows the non-obtrusive monitoring of physiological parameters and from which it might be possible to infer about affective states.”

3.

Sentence lines 40-41 - unclear, check sentence syntax or grammar?

The sentence in line 40-41 has been modified as follows:

“Therefore, it would be desirable that robots, which are designed to interact with adults and children, would be able to socially interact.”

4.

line 46 - to this purpose --> for this purpose

The authors thank the reviewer for finding the mistakes, that were corrected.

“ For this purpose, social robots should be able to express, through their shapes and behaviors, a certain degree of “intelligence” [5].

5.

line 85-86

", they may deliver much more reliable data about the emotional processes than visual channels. " - they deliver data on emotional SYMPTOMS, not processes

The sentence has been modified as follows:

“Therefore, they may deliver much more reliable data about the emotional symptoms than visual channels.”

6.

There is no need to distinguish section 1.1 if there is no section 1.2

Section 1.1 has been changed to section 2. According to the reviewer’s suggestion sections 3.1 and 6.1 have been also changed and turned into sections 5 and 9 respectively. All the other sections’ numbers have been changed accordingly.

7.

Overtime --> over time

(2 or 3 times throughout the paper, both are grammatically correct, but 'overtime' is extra hours (a noun) and 'over time' is an adverb expression meaning 'gradually in time', perhaps that what you meant)

The authors thank the reviewer for finding the mistakes, that were corrected in the following 2 sentences (line 257 and line 634).

“Vasomotor processes can be identified and monitored over time because they produce a thermal variation of the skin and they can be characterized by simple metrics such as temperature difference between data at two temporal points”

“These phenomena can be identified and monitored over time because they produce a thermal variation on the skin”

Reviewer 3 Report

The authors have addressed all issues raised in the previous review round.

Author Response

The authors thank the reviewer for the feedback.